# Research-based occupational therapy education: An exploration of students' and faculty members' experiences and perceptions

Kjersti Velde Helgøy[1]*, Jens-Christian Smeby[2], Tore Bonsaksen[3,4], Nina Rydland Olsen[5]

**1** Center of Diakonia and Professional Practice, VID Specialized University, Stavanger, Norway, **2** Centre for the Study of Professions, OsloMet – Oslo Metropolitan University, Oslo, Norway, **3** Department of Health and Nursing Science, Faculty of Social and Health Sciences, Inland Norway University of Applied Science, Elverum, Norway, **4** Faculty of Health Studies, VID Specialized University, Sandnes, Norway, **5** Department of Health and Functioning, Faculty of Health and Social Sciences, Western Norway University of Applied Sciences, Bergen, Norway

* kjersti.velde.helgoy@vid.no

**Data Availability Statement:** In accordance with restrictions imposed by the Norwegian Data Protection Services (NSD) with ID number 8453764, data must be stored on a secure server

## Abstract

### Introduction

One argument for introducing research in bachelor's degree in health care is to ensure the quality of future health care delivery. The requirements for research-based education have increased, and research on how research-based education is experienced is limited, especially in bachelor health care education programmes. The aim of this study was to explore how occupational therapy students and faculty members experienced and perceived research-based education.

### Methods

This qualitative, interpretative description consisted of three focus group interviews with occupational therapy students in their final year (n = 8, 6 and 4), and three focus group interviews with faculty members affiliated with occupational therapy programmes in Norway (n = 5, 2 and 5). Interviewing both students and faculty members enabled us to explore the differences in their experiences and perceptions.

### Results

Five integrative themes emerged from the analysis: "introducing research early", "setting higher expectations", "ensuring competence in research methods", "having role models" and "providing future best practice". Research was described as an important aspect of the occupational therapy bachelor program as it helps ensure that students achieve the necessary competence for offering future best practice. Students expressed a need to be introduced to research early in the program, and they preferred to have higher expectations

at VID Specialized University. The contents of the etics committe's approval resolution as well as the wording of participants' written consent do not render public data access possible. Access to the study's minimal and depersonalized data set may be requested by contacting the project manager, KVH, email: kjersti.velde.helgoy@vid.no or the institution: post@vid.no.

**Funding:** The author(s) received no soecific funding for this work.

**Competing interests:** The authors have declared that no competing interests exist.

regarding use of research. Competence in research methods and the importance of role models were also highlighted.

## Conclusions

Undergraduate health care students are expected to be competent in using research. Findings from our study demonstrated that the participants perceived the use of research during training as important to ensure future best practice. Increasing the focus on research in the programme's curricula and efforts to improve students' formal training in research-specific skills could be a starting point towards increased use of research in the occupational therapy profession.

## Introduction

Occupational therapists have positive attitudes towards research, but implement research evidence infrequently within their daily practice [1]. Professional education is believed to play an important role in the development of positive attitudes towards evidence-based practice (EBP) skills [2, 3]. One approach to improving evidence-based practice uptake in clinical practice is through the integration of research in education [4, 5]. Developing student's research skills is an important aspect of EBP [6] and participation in a research course has been found to improve nursing student's attitudes towards research [7]. The World Federation of Occupational Therapists recommends an occupation-focused curriculum that includes critical thinking, problem-solving, EBP, research and life-long learning [8, p. 6]. As such, educators in occupational therapy are advised to engage in research [8, p. 53].

Several studies have explored how to link research and teaching in higher education [9–20]. Based on previous research [9, 10, 15–20], Huet developed a research-based education model that distinguishes between research-led teaching and research-based teaching [14]. Research-led teaching means that academics use their expertise as active researchers or use the research of others to inform their teaching. Research-based teaching means that students develop research skills by being involved in research or other inquiry-based activities. Research-led teaching and research-based teaching is interconnected, and research and teaching should be seen as interlinked [14]. One strategy for linking research and teaching is to bring research into the classroom, e.g. through academics presenting their research relevant to the subject and discussing research outcomes and methods with students [14]. In a research-based learning environment, students learn how to become critical thinkers, lifelong learners and to generate discipline-enriching knowledge [14].

Research-based education has mainly been emphasized in disciplinary university education [21]. In medical education, students' knowledge, perceptions and attitudes towards research have been examined in several studies [22–25]. In their review, Chang and Ramnanan [22] found that medical students had positive attitudes towards research. Similar results emerged in Paudel et al.'s cross-sectional study [24]. Kandell and Vereijken et al. [25] found that first-year students believed that research would be important to keeping up to date in their future clinical practice.

Less research has been carried out on research-based education in bachelor's programmes in health care, but the requirements for research-based education have also increased in these programmes [21, p. 11]. Some studies have, however, investigated the attitudes, skills and use of research among nursing students [4, 7, 26]. In their literature reviews, Ross [7] found that

nursing students have positive attitudes towards research and Ryan [26] found that nursing students are generally positive towards the use of research. Ross [7] noted that participation in research courses and research-related activity improved students' attitudes towards nursing research. Leach et al. [4] have argued that undergraduate research education has an impact on nursing students' research skills and use of EBP.

Students in occupational therapy and physiotherapy have been found to share positive attitudes towards research [27]. Similarly, studies have found positive attitudes towards EBP among occupational therapy students [28–31]. Stronge and Cahill [29] found that students were willing to practice EBP, but a lack of time and clinical instructors not practising EBP were perceived as barriers. Stube and Jedlicka [28] and Jackson [30] highlighted the importance of learning EBP through fieldwork experiences. DeCleene Huber et al. [31] found that students were least confident in EBP skills that involved using statistical procedures and statistical tests to interpret study results. These studies focused mainly on students' attitudes and competence in EBP and research utilization (RU), and other elements of research-based education, such as student's exposure to and engagement with research evidence were not investigated.

Few studies have explored faculty members' perceptions of research in undergraduate education. Wilson et al. [32] found that the way in which university teachers in university disciplines translate research into learning experiences depends on their own personal perception of research. Some academics have highlighted that disciplinary content must be learned before engaging in research [32]. This idea is in accordance with the findings of Brew and Mantai [33], who also found variations in the way in which academics conceptualized undergraduate research. Experiences, attitudes and barriers towards research have been examined among the junior faculty of a medical university, and findings indicate that fewer than half of the participants in the study were involved in research at the time [34]. Ibn Auf et al. [35] found that the factors significantly influencing positive perceptions of research experience among the faculty at medical programmes were being male, having had education in research during undergraduate level, having been trained in research following graduation, and having undertaking years of research. To our knowledge, few studies have been conducted on faculty members' perceptions of research in health care education, and we have identified only one such study related to occupational therapy education [36]. In a survey Ordinetz [36] found that the faculty members had a positive attitude towards research-related activities, and they considered research as an integral component of their role. Still, participants found research-related activities difficult to perform.

Research into how research-based education is experienced and perceived by faculty members and students is limited, especially in bachelor's programmes in health care [21, p. 11]. To gain a better understanding of the advantages or disadvantages of linking teaching and research, we therefor aimed to explore students' and faculty members' experiences and perceptions of research-based education in one bachelor's programme in health care in Norway. Our specific research questions were:

1. *How do students and faculty members in occupational therapy programmes perceive the emphasis on research in the programme?*

2. *How do occupational therapy students and faculty members perceive the expectations regarding research during the programme?*

3. *What similarities and differences exist between the experiences and perceptions of students and faculty members regarding research-based education in the programme?*

## Context of the research

In Norway, it is required that higher education should be research-based [37]. According to the Act relating to universities and university college § 1–1 b, education must be on the cutting edge in terms of research, development work and artistic practice [37]. In a recent white paper on quality in higher education, research-based education is defined as education that is linked to a research environment; is conducted by staff who also carry out research; builds on existing research in a particular field; provides knowledge about the philosophy of science and research methods; and provides opportunities for students to learn how research is conducted from staff or students themselves conducting research as a part of their studies [38].

In Norway, six higher education institutions offer bachelor's programmes in occupational therapy. The bachelor's degree in occupational therapy consists of 180 European Credit Transfer and Accumulation System (ECTS) credits and covers four main areas of learning: the natural sciences, medicine, the humanities and the social sciences as described in the National Curriculum [39]. In total, clinical education consists of 60 ECTS. Philosophy of science and research methods comprise 6 ECTS, and research and development consist of 9 ECTS. The bachelor project module involves research and development work in occupational therapy and consists of 15–20 ECTS, with some variation across programmes. According to the national curriculum, the purpose of the occupational therapy education is to prepare students to be professionally up-to-date, future-oriented and research-based in their practice [39]. Students should be able to read research papers and use research results in their professional practice, and they should be able to justify their actions to users, other professionals and employers. These regulations have typically been operationalized as learning outcomes, such as: "Students will be able to apply relevant theories and research results, to understand people with activity problems as a result of somatic illness or injury, and to be able to make reasoned choices of intervention", or "Students will be able to find and apply research to justify an intervention in occupational health" [40].

Most higher education institutions in Norway require that faculty members hold a master's degree and many require a PhD or equivalent. It is regulated by law that at least 20% of faculty members in programmes at the bachelor's level have PhD or equivalent according to the regulation concerning supervision of the quality of education in higher education § 2–3 (4) [41]. The amount of time that faculty members are able to use on research and development varies between institutions. The type of academic position will often influence the percentage of time for which they can apply; 20% of a full-time position for assistant professors, 30% of a full-time position for associate professors and 40% of a full-time position for full professors [42].

## Methods

We used a qualitative design with empirical data from six focus group interviews, and the research strategy 'Interpretative description' guided the analysis. Interpretive description is an inductive approach inspired by ethnography, grounded theory and phenomenology [43], and is a research strategy suitable for studying phenomena in practical disciplines, such as nursing, teaching and management [44]. It is an approach driven by a fundamental belief in the rightness of striving to find better ways to serve one's disciplinary purpose, and the potential of research to guide one there [43, p. 12].

## Participants

A purposive sample was recruited consisting of two different participant categories: final year occupational therapy students and faculty members. Participants were recruited from three different bachelor's programmes in Norway. We contacted programme managers, who helped

us recruit faculty members. We used the students' digital learning platform to inform them about the study and students were encouraged to contact us via email should they wish to participate in the study. Focus group interviews with students were conducted after their final clinical placement, when they had just started working on their bachelor's projects. The study sample for this project consisted of two groups: third-year occupational therapy students (n = 18), and faculty members (n = 12). In total, 30 persons between the ages of 21 and 64 participated in the study (Table 1), 7 of whom were men.

## Data collection

In total, we conducted six focus group interviews with occupational therapy students and faculty members from three different bachelor's programmes. All focus groups were conducted during the spring of 2019. The focus groups with students and faculty members were conducted separately, due to differences in expertise and power; according to Krueger [45, p. 22], this is necessary to ensure that participants feel free to speak openly. The size of the focus groups ranged from two to eight participants (Table 1), and they lasted between 76 and 86 minutes. A digital voice recorder was used to audiotape each session. The focus groups were held at a convenient time and location for the participants, such as meeting rooms on campus. All participants were asked to provide written consent before participating in the study, and completed a form, giving brief details about their background. A thematic interview guide was

**Table 1. Characteristics of the participants.**

| Characteristics | Students | Faculty members |
|---|---|---|
| Invited/participated | 145/18 | 30/12 |
| Number of focus group interviews | 3 | 3 |
| Sex | | |
| Men | 6 | 1 |
| Women | 12 | 11 |
| Age | | |
| 20–29 | 15 | 1 |
| 30–39 | 3 | 3 |
| 40–49 | | 2 |
| < 50 | | 6 |
| Clinical experience (years) | | |
| 0–4 | 10 | 2 |
| 5–9 | 3 | 2 |
| 10–19 | | 6 |
| < 20 | | 2 |
| Teaching experience (years) | | |
| 0–4 | | 5 |
| 5–9 | | 1 |
| 10–19 | | 4 |
| < 20 | | 2 |
| Highest degree obtained | | |
| Bachelor's degree | 2* | |
| Master's degree | | 8 |
| PhD or equivalent | | 3 |

*Held a bachelor's degree in other subject area before entering the bachelor's program in occupational therapy.

developed, based on the aim of this study and on previous research on research-based education. The following topics were covered: 1) perceptions of the term "research-based education"; 2) expectations regarding the use of research in education, including clinical placements; 3) students' involvement in research projects and the faculty's experiences regarding such involvement; 4) faculty' members participation in research; 5) experiences with teaching research methods and the philosophy of science; and 6) research-based education and future professional practice. The first author developed the interview guide, drawing on previous research on the subject, and in cooperation with the three other authors. The authors are faculty members at different educational institutions. All authors were interested in the topic of research-based education, and were experienced both in teaching and supervising health care students, and conducting focus groups.

## Ethical considerations

The Norwegian Data Protection Services (NSD) approved the study (ID number 845364). Participation in the study was voluntary and the participants had the opportunity to withdraw from the study at any time without consequences. All transcripts and notes were anonymous and written consent was obtained from all the participants. None of the authors conducted focus group interviews with their own colleagues.

## Analysis

Previous research into research-based education served as an important starting point for the interview guide and the analysis. Interpretive description guided the process of analysing the data [43]. The co-moderators took notes during the focus group interviews, and all focus group interviews were transcribed verbatim. After each focus group interview, the moderator and co-moderator engaged in a short debriefing session. The first and last author (KVH and NRO) performed the analyses, separately at first, followed by a joint analysis in which the two authors discussed and compared their interpretation of the data and agreed on patterns and themes. Word processing was used to analyse the data, and the analysis consisted of a series of operations: 1) reading the transcripts many times while being as open-minded as possible; 2) writing marginal remarks by consistently questioning the text and pointing out important points, potential themes or patterns; 3) condensing; 4) broad coding, 5) comparing and contrasting within focus groups with similar participant categories; and finally 6) comparing and contrasting focus groups compromised of different participant-categories. Comparing and contrasting within and between focus group interviews, enabled us to generate patterns and themes within the entire data set.

The analysis was characterized by a back and forth process that involved taking things apart and putting them back together again. Throughout this process, the first author frequently returned to the transcripts to ensure that the interpretations reflected the data. To ensure rigor and credibility in the analysis, the authors stepped away from the data periodically to ask questions such as: "what am I seeing?", "why am I seeing that?", "how else might I understand this aspect of data?", "what might I not be seeing?" and "what are they not telling me?" [43, p.174]. This approach prompted the authors to see the data through "alternative lenses" and to acknowledge that there was much else to be seen [43, p. 174].

## Results

The aim of this study was to explore how occupational therapy students and faculty members experienced and perceived research-based education. Across the focus group interviews, we identified five integrative themes; "introducing research early"; "setting higher expectations";

"ensuring competence in research methods"; "having role models", and "providing future best practice". Students felt it was important to be introduced to research early on in the programme, meet high expectations regarding the use of research, gain competence in research methods and have role models who use research evidence during clinical education. Faculty members felt that students needed competence in research methods and they highlighted the importance of linking research to professional practice. We found the most contrasting views between students and faculty members to be related to the theme "setting higher expectations". While the students would have liked more focus on research, the faculty members discussed whether expectations regarding the use of research were too high. Both students and faculty members across the focus group interviews believed that research-based education was important with regards to helping students achieve the necessary competences to provide future best practice.

## Introducing research early

Across the focus group interviews, several students expressed the belief that focusing on research early in their education would better enable them to read and understand research during their training. Furthermore, one participant highlighted that occupational therapists are expected to integrate up-to-date research findings in their practice, and learning about research in the early stages was important in order to ensure this competence. Most students questioned the timing of when research should be introduced to students during their training. One of these students was of the opinion that learning about research should be mandatory from the start of the programme:

> "I wished that teaching related to research was mandatory early in my studies. Everything you want to be good at requires practice, right? I wished that I had better research skills; [in particular concerning] searching for and using articles before we started working on our bachelor's project."
>
> (Student, Focus group 3)

This participant had also experienced benefits, such as higher grades, when using research articles in assignments. One of the other students highlighted that critical thinking is also part of research-based education. Students expressed a need to be critical with regard to research, and several of them wished that the curriculum had focused on how to read an article critically. Some of the participants pointed out that the use of research was not introduced until their final year, which they felt was too late.

Learning about research early on in the programme was also mentioned in one of the focus group interviews with faculty members. Some of the faculty members believed that learning to use research should be introduced at the start of the programme, as research is a natural part of academic development. For students to be able to read research articles early, they must also learn about research methods early on in the programme.

> "I think it is important for [students'] academic development to start early. That they start early, and really expects that they will implement it in their first, second and third year, and that it won't just come abruptly in third year."
>
> (Faculty member, Focus group 6)

In contrast to this, one of the faculty members expressed that early on, some students seem more focused on understanding what it will be like to practice occupational therapy, rather than learning about research evidence:

"Of course, one must start [focusing on research], but I understand students who wish to understand what the profession is like first and foremost, right? What will my day consist of as a clinician when I graduate, that is what they want to know first, right?"

(Faculty member, Focus group 6)

This faculty member felt that first-year students strived to grasp the concept of occupational therapy, and that understanding the profession is essential early on in their training. Later, students can develop their understanding and use of research evidence.

## Setting higher expectations

Several students expressed that they would have preferred their teachers to have set higher expectations early on regarding the use of research during their training. They had the impression that the expectations had changed for new student cohorts, and they wished that they had been given the same opportunity. In contrast, however, one student described having been given clear expectations regarding the use of research articles during the first year, although these expectations decreased as the program progressed. This participant felt that the expectations regarding the use of research in assignments were too low, especially after the first year:

"It. . . seems sufficient to include a sentence from an article, and to refer to one research article in the reference list, and then it is okay in a way. I think the expectations are too low. We must include a research article, but why do we do this really? What is the point? Nothing more is required."

(Student, Focus group 2)

Some students also experienced that the expectations regarding the use of research use were too low during their clinical placements. They expressed that it was unusual to talk about theory and research during placement. As future clinical instructors, they would expect more research use from their students.

We found the most contrasting views between students and faculty members regarding this theme. While students would have liked more focus on research, the faculty members discussed whether expectations regarding the use of research were too high. In one focus group with faculty members, participants discussed how students' motivations regarding the use of research varied. In their experience some of the students were mainly interested in hands-on practice, whereas others were more interested in research. In another focus group, faculty members indicated that they were satisfied with the requirements and expectations regarding research use among their students. They highlighted that learning about research requires maturation, and that the extent to which research can be integrated during training is limited.

Another factor that was highlighted by faculty members was a fear that the increased demand for research-based education could create a distance to professional practice. Some participants highlighted that it takes time to understand the profession and that too much emphasis on research in education could threaten this process. In one focus group interview, there was a discussion regarding how to balance the emphasis on research and profession- specific knowledge:

One participant remarked that research was too much in focus:

"In light of the national expectations regarding research-based and evidence-based practice, we need to ask ourselves, what is our profession, right? We live in a time and a society where evidence-based practice and research is almost emphasized too much."

(Faculty member, Focus group 6)

In contrast, however, one of the other faculty members in the same focus group, stated that one does not forget the importance of the occupational therapy profession when conducting research, and emphasized the importance of research in education.

In another focus group interview, some of the faculty members were concerned about the increasing demand for research competence among faculty members. They feared a situation where the majority of the faculty might have research competence, but limited clinical experience, which might affect the students' learning of specific occupational therapy skills:

"Occasionally I fear that we have less of the experience-based knowledge in the faculty, now that faculty members are expected to have a PhD. Many [faculty members] start their career at the university colleges and universities and complete the doctoral degree, without having much clinical experience."

(Faculty member, Focus group 1)

## Ensuring competence in research methods

Both students and faculty members highlighted that student's needed competence in research methods. The students emphasized the need for more competence and skills regarding quantitative methods and statistics, to enable them to read and understand research articles. Several students stated that, while they had some knowledge about qualitative methods, they had not learned much about quantitative methods. The importance of competence in research methods was highlighted in one of the focus group interviews with students:

"If you have competence in [research] methods, then it might be easier to read and understand a research article. Often when I try to read a research article about something medical, I skip the methods section because I don't understand it."

(Student, Focus group 2)

Students believed that reading research articles prepared them for clinical work. In fact, one participant was very aware of a lack of competence in research methods during a clinical placement and expressed this as follows:

"I experienced that clinical instructor said in the beginning of the placement "you [as a student] can use research, as you are good at it". "This is perhaps a bit unfortunate, since we haven't been sufficiently introduced to how to find and interpret research, and not early enough. As such, we didn't used it [research] in the placement, because we haven't been confident enough."

(Student, Focus group 3)

Competence in research methods was also discussed in relation to students' bachelor's projects. Students expressed that it was too late to start learning about research methods in the final year. If research methods were not introduced at an earlier stage, students felt unprepared

for their bachelor's project. Students stated that expectations regarding research use only took place in one exam, and they wished that they had been challenged more regarding the use of research throughout the programme. Benefits from reading research articles compared with regular textbooks were highlighted, such as learning more about the results from interventions within different patient groups. Faculty members also felt that teaching methods and the philosophy of science was not sufficiently integrated in the programme. They noted that a consequence of this was a lack of competence in research methods among students, which in turn was a challenge regarding student participation in research projects initiated by faculty members. As one participant explained:

> "What I think now is that of course they have received too little teaching related to research methods to be able to contribute. I have a bad conscience that they have received too little basic knowledge to get started doing it."

(Faculty member, Focus group 5)

## Having role models

Some of the students experienced working alongside role models who used research evidence, both on campus and in clinical placements. With regards to clinical instructors acting as role models, one of the students stated:

> "The clinical instructors used a lot of research, so I felt that I also had to [use research evidence] to keep on track."

(Student, Focus group 2)

A student from another focus group also experienced clinical instructors as expecting students to use research, although this varied from placement to placement:

> "If I had treated a patient, I had to justify this with research, but it was mostly when I had placements in hospitals, so it depended very much on the clinical placement site."

(Student, Focus group 4)

Some of the students encountered the expectation that nearly all treatments performed during their placements should be justified with research evidence, especially in hospital-based services. For example one particular student described:

> "In my final clinical placement in occupational health services, it was only research that mattered. At this placement you were expected not to say anything without reading up on legislation and the latest knowledge."

(Student, Focus group 4)

In contrast to this, however, several other participants had the impression that their clinical instructors did not use much research but instead emphasized experienced-based knowledge and placed more emphasis on the students' interaction skills. These students perceived that their clinical instructor's decision-making was based on experience-based knowledge. One participant experienced this lack of emphasis on research in clinical placements as follows:

"My clinical instructor in my last clinical placement, started to say "I am probably not going to test you on use of research evidence, because there you are superior to me, since it has been so long since I have done that. Rather, I will keep an eye on your communication skills and how you appear."

(Student, Focus group 2)

Later in the focus group interview, the same participant stated:

"I wasn't surprised, because this experience was similar to earlier experiences from other clinical placements: that we are not challenged on research-based stuff by the clinical instructor."

(Student, Focus group 2)

This is similar to the view of another participant who expressed that research use was not visible in any of the clinical placements. Despite or perhaps because of this experience, this participant saw the need to use research to keep-up-to date in future clinical practice.

Faculty members across the focus group interviews noted a variation regarding clinical instructors' engagement in research evidence; however, one participant highlighted an increasing focus on research among clinicians:

"I always address [research] during clinical placement visits when clinical instructors are present, and I experience that the practice field is much more focused on research now than before. Many say that this is thanks to the students."

(Faculty member, Focus group 1)

As such, students could also influence the use of research evidence during clinical education. One of the faculty members confirmed this when noting that the students requested justifications based on research after lectures on making orthoses. This faculty member recalled that students had stated that they wished the clinician lecturer had used research evidence to better justify this type of intervention. The faculty member believed that students developed their critical thinking skills during the programme and became increasingly interested in how research supported clinical decisions. Some of the faculty members highlighted that they made an effort to stimulate students' critical thinking skills, and that they were focused on educating students that could integrate critical thinking into their professional practice.

Students also experienced faculty members as role models in terms of using research in various ways. Some of the students described a difference between younger and more experienced faculty members. They found that younger faculty members who held a PhD and were recently hired, were more focused on research than faculty members with more clinical experience. The students described this as a generational shift that has possible led to an increased focus on research in the programme recently. Overall, the students' impression of their teachers as active researchers varied. Some students had the impression that some of their teachers did not conduct much research, and others had the impression that all of the faculty members conducted research, that they invited students to participate in their projects, and that their research was visible both in their teaching and in reading lists:

"They talk about their research projects, so everyone as far as I know is involved in research in some way. I also see their names on research articles."

(Student, Focus group 3)

Furthermore, none of the students in the three focus group interviews remembered that teachers had focused on research during students' clinical placements visits:

"I didn't get any questions about research from the teacher during my placement visit. In a way I feel that everybody says that research is so important, but it feels like it is mostly experience-based."

(Student, Focus group 2)

In one of the focus group interviews with students, some participants described their uncertainly regarding whether their teachers were basing their lectures on research, as the Power Point presentations often lacked references. One student remarked:

"Since [the teachers] refer to theory using old sources, I don't trust that they have searched for new research."

(Student, Focus group 3)

## Providing future best practice

Across the focus group interviews, participants perceived that research in occupational therapy bachelor's programmes was important with regard to helping students achieve the necessary competence to provide future best practice. Both students and faculty members expressed a need to use research as an information source to justify their professional practice to other professions and collaborators. From the students' perspective, research can provide professional credibility and drive the profession forward. One student expressed the following:

"You. . . desire that the intervention or the treatment or what you provide is the most professionally credible option, and that this option will have the best possible effect."

(Student, Focus group 2)

This participant described the importance of incorporating research in training and clinical placements, to ensure that graduates integrate research in future clinical practice. Furthermore, the participant expressed a responsibility to keep-up-to date with research. One of the other participants in this focus group agreed, and emphasized that when working with patients, it is important to be familiar with the latest research evidence in a particular field.

Faculty members emphasized that graduates should have the necessary competence to be able to justify treatment choices to meet the demands of society. One of the faculty members stated that emphasis on research in the programme would probably increase the focus on research after graduation, when students would be less exposed to research than they were during their training. Another stated explicitly that being able to justify treatments using research, as opposed to using intuition only, could empower students in their future practice:

"When you work in hospital-based services, if you suggest an intervention, then you will be asked why you would do that and then you can justify it with research. And that is something I believe, the benefits of empowering our students to meet the demands of society, especially in hospital-based services."

(Faculty member, Focus group 1)

## Discussion

Our aim was to explore how occupational therapy students and faculty members experienced and perceived research-based education. We found that students were engaged in learning about research and they considered research to be important. Both students and faculty members perceived that research in bachelor's programmes was important with regards to helping students achieve the necessary competence to provide future best practice in occupational therapy. Students expressed a need to be introduced to research in the early stages of the programme, and for higher expectations regarding the use of research during their education. Both students and faculty members acknowledged the need for students to gain competence in research methods, as this would enable them to read research articles and participate in research projects. Students maintained that both clinical instructors and faculty members were important role models in the use of research evidence.

In this study, students expressed that they would have preferred to learn about research earlier in their programme, and some of the faculty members highlighted that integrating research at an early stage was important for academic development. Early integration of research and enquiry has also been emphasized by Healey and Jenkins [15] and Jenkins and Healey [46]. Jenkins and Healey [46] have suggested that institutions and departments develop courses that engage students in research and enquiry from the beginning of their first year. They used the term "enquiry" to highlight the importance of curiosity, as well as critical thinking. An argument for early integration of research and enquiry is to enhance the linkage between teaching and discipline-based research [15]. Walkington et al. [47] have argued that all undergraduate geography students would benefit from early attempts to develop skills in enquiry and research. They found that students felt more prepared to undertake research independently when they were given the opportunity to practise research skills in advance [47]. This is relevant to our study findings, as the bachelor's project in the final year in the occupational therapy programmes involves research-related tasks, but learning research methods takes time. Accordingly the students who participated in our study reported that they would have liked to learn about research methods earlier, as this would have better prepared them for undertaking their bachelor's project.

Some of the faculty members in our study believed an early introduction to the use of research is a part of academic development and the importance of progressing in this area throughout the programme was underlined. In contrast, however, one of the faculty members expressed that, in the early stages of the programme, some students seemed to be more focused on understanding the role of the occupational therapist than learning about research. This contrasting view coincides with the findings of Wilson et al. [32], who found that some academics appear to have a hierarchical understanding of research, in which disciplinary content must be learned before engaging in research. Nevertheless, students and faculty member alike explicitly articulated the need for students to develop and learn critical thinking skills at an early stage in their programme, such as how to read an article critically. Critical thinking skills were highlighted as being both important for professional practice and research-based education. However, integrating the use of research evidence early also comes with challenges. Kyvik et al. [48] found that undergraduate professional students developed only limited understanding of research. This speak to the importance of integrating the use of research as early as possible in students' training, as these are competences that students need in order to engage in evidence-based practice as professionals.

Most of the students experienced a lack of expectations regarding the use of research in assignments and during clinical placements, whereas some of the faculty members wondered whether the expectations were too high. Some faculty members feared that the increased

demand for research-based education could widen the gap between education and professional practice. Interestingly, it appears that engaging in research and professional practice may be perceived as two separate domains, with some ambiguity concerning the role of research in the occupational therapy profession. This provides further support to Kyvik's [21, p. 142] argument that research-based education in bachelor programmes in health care should place an emphasis not just on enquiry-based learning but its relevance for professional practice.

Both students and faculty members in our study described competence in research methods among students as important. Students reported a need for competence in research methods, including quantitative methods, to be able to understand research articles. Moreover, faculty members reported that the lack of students' competence in research methods could represent a challenge when including students in their own research projects. However, Decleene Huber et al. [31] found that students lacked confidence in using statistical procedures and statistical tests to interpret study results. Kyvik et al. [48] and Brew and Mantai [33] found that undergraduates lacked the research skills needed to be involved in faculty projects. This represent a potential challenge, as most students in our study reported that they wanting to know more about their teachers' research and to be involved in their research projects. Students also highlighted that involvement in faculty research might be a way to inspire them to continue with research in the future. Healey and Jenkins [15] have argued that students seemed more motivated if they were integrated in their teachers' research projects at an early stage in their studies. Moreover, Smyth et al. [49] claimed that students value research experience and experienced benefits from engaging in research such as an improved understanding of the research process, increased critical thinking and professional and practical skills. This supports the view that learning about research methods, including quantitative and statistical methods, should be integrated throughout the programme.

In our study, students highlighted the importance of role models with regards to using research, especially during clinical placements. Students reported that they were positively influenced by clinical instructors who used research in their work, and as a result, they felt obliged to use research to maintain their own professional development. Students who had experienced the opposite still reported that they had searched for and used research evidence to keep-up-to date. The importance of role models in promoting EBP and the use of research evidence has been reported in several studies [28–30, 50, 51]. Olsen et al. [50] and McCluskey [51] found that clinical instructors were important role models regarding EBP for students in clinical placements. Stube and Jedlicka [28] have suggested that educators have a role in assisting students to become scholarly consumers of evidence. In our study, most of the students had the overall impression that faculty members used research in their teaching, but the degree to which students were involved in research projects varied. In one focus group interview, students highlighted that all the faculty members were active researchers and published articles. However, students found that faculty members visiting students during placements were less inclined to focus on research. Some students also reported a focus on experienced-based knowledge more than on research-based knowledge among some clinical instructors. Closer cooperation between education and practice might be a way to increase the emphasis on research during clinical placements. Role models for use of research is of importance.

Experience with research and development work during training may enhance interest in applying research in one's future working life [48]. Future quality health care delivery is often a main argument for integrating research in bachelors' programme in health care education, such as occupational therapy education. In our study, both faculty members and students highlighted the importance of focusing on research in training to ensure that graduates have the necessary competence to provide future best practice. Previous research findings also

indicated that students have a strong belief in the value of research for their future clinical practice [25] and desire to keep up-to-date in their field [23]. To achieve this, it is important that students start as early as possible and ideally during their formal education [2]. Deicke et al. [52] have highlighted that students need to work with actual research literature, develop research designs and undertake empirical research to increase their interest in research. Findings from our study indicate that emphasis should be placed on introducing the use of research at an early stage in the programme and higher expectations should be set for students regarding the use of research, both on campus and during clinical placements, to ensure that students achieve competence in research methods and that they are exposed to role models who use research evidence.

Occupational therapists have positive attitudes towards research, but infrequently implement research evidence in their daily practice [1]. This may be due to barriers such as lack of time to read research, insufficient facilities and difficulty understanding statistical analyses [53]. Results from our study indicate that a link between research and profession-specific knowledge is necessary for research-based education in bachelor's programmes in health care education. Research should be integrated in teaching. Along this lines, Huet [14] has argued that faculty members need to be engaged in a scholarly manner within their disciplinary field, and has highlighted the importance of raising a culture of research and teaching as two integrated activities. Learning in a research-based education environment may ensure that students enriches the knowledge in their discipline [14]. Students need to be socialized into a culture of research. Ideally, the learning of research-specific skills should be integrated into all fields of learning in a programmes curricula and include skills in critical thinking, problem-solving, EBP and research, as emphasized by WFOT [8]. Increasing focus on research in curricula and improving the students' formal training in research-specific skills could be a starting point towards increased use of research in the occupational therapy profession.

## Limitations

This study explored the experiences and perceptions of occupational therapy students and faculty members regarding research-based education in three different occupational therapy programs in Norway. Conducting focus group interviews with both students and faculty members enabled us to explore and compare differences in perceptions and experiences. Focus groups compared to individual interviews can potentially create a synergy that is not possible in individual interviews [54, p. 18]. The challenge with this method, is that there is a possibility that dominant participants can influence the results and participants tend to intellectualize [45, p. 22, 13]. In our study, we did not experience dominant participants, although one of the participants in one of the focus group interviews contributed very little. We made efforts to include this participant more directly in the conversation.

This study was conducted in the early phase of third-year students' bachelor's projects. There is a possibility that the students were more focused on research during this phase of their training. It is also possibly that students who participated in this study could have been more interested in research-based education than students who chose not to participate. While a specific interest in research-based education may also have been the case for the faculty members who chose to participate in the study, we note that only three of the participating faculty members held a PhD or equivalent, and the majority of their experience originated from clinical practice. Thus, when considering the faculty's views as expressed in this study, the characteristics of the group should be considered. In addition, one of the focus groups, made up of faculty members consisted of only two participants. A strenght of our study, is that two authors conducted the analysis. We considered using member checking, but decided against it, as,

according to Thorne [43 p. 175], member checking can lead to false confidence if the participants confirm what you thought or potentially derail you from good analytic interpretations if they do not. With regards to the participants, however, we feel that including both students and faculty members as study participants represents a strength of the study. A corresponding limitations is that we are unable to assess the extent to which the study sample is representative of the population of faculty and students in occupational therapy education in Norway. However, as establishing representativity is generally not an aim of qualitative studies, and according to Thorne [43, p. 105], there is no definitive rule regarding the correct sample size for an interpretive description study, we believe our findings still offer important insights for the field.

## Conclusions

This study explored the experiences and perceptions of occupational therapy students and faculty members from three occupational therapy bachelor's programmes in Norway regarding research-based education. Students in these programmes are expected to be competent in using research evidence. Findings from our study show that both students and faculty members perceive the use of research during training as important in order for students to provide future best practice. Furthermore, findings indicate that setting high expectations regarding the use of research early on, may be important in a bachelor's programme in health care such as occupational therapy. Ensuring competence in research methods seems to be essential for achieving success in terms of research-based education. Moreover, clinical instructors and faculty members were highlighted as important role models in the use of research evidence. Future research is needed that focuses on the use of research by clinical instructors and their expectations of students regarding using research in clinical placements. Quantitative studies are also needed, so that a wider population can be reached, as are more focus groups among students and faculty members in other professional education programmes.

## Supporting information

**S1 File. Interview guide for faculty members in English.**
(DOCX)

**S2 File. Interview guide for faculty members in Norwegian.**
(DOCX)

**S3 File. Interview guide for students in English.**
(DOCX)

**S4 File. Interview guide students in Norwegian.**
(DOCX)

**S1 Checklist. ISSM COREQ.**
(PDF)

## Author Contributions

**Conceptualization:** Kjersti Velde Helgøy, Jens-Christian Smeby, Tore Bonsaksen, Nina Rydland Olsen.

**Formal analysis:** Kjersti Velde Helgøy, Nina Rydland Olsen.

**Investigation:** Kjersti Velde Helgøy, Jens-Christian Smeby, Tore Bonsaksen, Nina Rydland Olsen.

**Methodology:** Kjersti Velde Helgøy, Jens-Christian Smeby, Tore Bonsaksen, Nina Rydland Olsen.

**Project administration:** Kjersti Velde Helgøy, Tore Bonsaksen, Nina Rydland Olsen.

**Supervision:** Jens-Christian Smeby.

**Validation:** Kjersti Velde Helgøy.

**Writing – original draft:** Kjersti Velde Helgøy.

**Writing – review & editing:** Jens-Christian Smeby, Tore Bonsaksen, Nina Rydland Olsen.

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
