## [Decision Letter · Decision Letter 0]

22 Jul 2020

PONE-D-20-09978

Research-based occupational therapy education: an exploration of students` and faculty members` experiences and perceptions

PLOS ONE

Dear Dr. Helgøy,

Thank you for submitting your manuscript to PLOS ONE. After careful consideration, we feel that it has merit but does not fully meet PLOS ONE’s publication criteria as it currently stands. Therefore, we invite you to submit a revised version of the manuscript that addresses the points raised during the review process.

All reviewers pointed to main strengths of the manuscript. But they identified issues, especially at the conceptual and methodological level, that require major revisions.

We look forward to receiving your revised manuscript.

Kind regards,

Sara Rubinelli

Academic Editor

PLOS ONE

Journal Requirements:

2. Please include a copy of Table 2 which you refer to in your text on page 14

Reviewers' comments:

Reviewer's Responses to Questions

**Comments to the Author**

1. Is the manuscript technically sound, and do the data support the conclusions?

Reviewer #1: Yes

Reviewer #2: Yes

Reviewer #3: Yes

2. Has the statistical analysis been performed appropriately and rigorously? 

Reviewer #1: N/A

Reviewer #2: N/A

Reviewer #3: N/A

3. Have the authors made all data underlying the findings in their manuscript fully available?

Reviewer #1: Yes

Reviewer #2: Yes

Reviewer #3: Yes

4. Is the manuscript presented in an intelligible fashion and written in standard English?

Reviewer #1: Yes

Reviewer #2: Yes

Reviewer #3: Yes

5. Review Comments to the Author

Reviewer #1: Thank you for the opportunity to review this paper, which explores students’ and faculty members’ experiences with and perceptions of research-based occupational therapy education. The study method was described in detail and the analysis was conducted in line with the interpretative description methodology. I particularly enjoyed reading the quotes, which further demonstrated the depths of the research finding.

I have some concern with the way that ‘research-based education’ is conceptualised in the manuscript. Research-based teaching refers to teaching practice “…when academics plan, deliver and assess students’ work through their involvement in research or inquiry-based activities” (p.728, Huet, 2018 – Research-based education as a model to change the teaching and learning environment in STEM disciplines, published in the European Journal of Engineering Education, 43:5, 725-740). A similar term – ‘research-lead teaching’ – occurs when academics use their expertise as active researcher or use the research of others to inform teaching. Without knowing how the occupational therapy curriculum is structured, I find it difficult to accept that it is indeed research based. It would have been beneficial for the authors to give more detail about how the curriculum is designed and delivered and to provide a few examples of research-based learning activities. Why is occupational therapy education considered as a short professional education programme, given that it is a bachelor degree? What are the expectations placed on the lecturers teaching in the occupation therapy programme, in terms of their engagement with research? Are they all clinicians? Again, providing more information about the course structure might help reduce the confusion here.

I would have liked to see more rationale given to using interpretive description as the study methodology. What’s the advance of this approach and why is it most suitable for the current study?

Participants commented on the fear that “… the increased demand for research-based education could cause a distance to professional practice”. It would be important to discuss whether comments like this fits the overall research-based model and to provide solutions to reduce such fears. I personally think that research and teaching should not be competing against each other and that the whole premise of research-based education is to incorporate research into teaching, as a means to facilitate active inquiry and critical thinking.

Reviewer #2: 1) 1. The qualitative research in the care activities is interesting, related to the management of good health science education in explaining the views, phenomena and perceptions of both the instructor and the learner.

2. The method should be improved to give details of the research methodology to understand more clearly how to do, what to do and how to analyze in order to get reliable research results Due to a small sample number

3) 3) Summary of themes is good but should make the diagram interesting In order for readers to understand the elements discovered in quality research more clearly, will make this research suitable for dissemination that useful in occupational therapy for the publication of world-renowned journals. Should adjust the diagram to be clear

In my opinion, this paper is good for occuaptional therapy education and health science education. It can share in PLoS ONE journal to contribute around the world. I’m Ok accpet thay Minor Revision,

Best Regards

Supat Chupradit, Ph.D.

Assistant Professor in Occupational Therapy, Depaertment of Occupational Therapy, Faculty of Associated Medical Sciences, Chiang Mai University, Thailand

Reviewer #3: Exploring the perceptions of OT students and faculty regarding their experiences with receiving education about research is an important topic related to encouraging evidence-based practice in the field of OT. See attached review for detailed comments and recommendations for revision.

6. PLOS authors have the option to publish the peer review history of their article (what does this mean?). If published, this will include your full peer review and any attached files.

Reviewer #1: **Yes: **Yan Chen

Reviewer #2: **Yes: **Supat Chupradit

Reviewer #3: No

---

## [Author Response · Author response to Decision Letter 0]

29 Oct 2020

Dear Editor and Reviewers: I have incorporated all of your suggestions into my revision. They were very helpful. Thank you. Please find them adressed in a separate document.

---

## [Decision Letter · Decision Letter 1]

24 Nov 2020

Research-based occupational therapy education: an exploration of students` and faculty members` experiences and perceptions

PONE-D-20-09978R1

Dear Dr. Helgøy,

We’re pleased to inform you that your manuscript has been judged scientifically suitable for publication and will be formally accepted for publication once it meets all outstanding technical requirements.

Kind regards,

Sara Rubinelli

Academic Editor

PLOS ONE

Additional Editor Comments (optional):

Reviewers' comments:

Reviewer's Responses to Questions

**Comments to the Author**

1. If the authors have adequately addressed your comments raised in a previous round of review and you feel that this manuscript is now acceptable for publication, you may indicate that here to bypass the “Comments to the Author” section, enter your conflict of interest statement in the “Confidential to Editor” section, and submit your "Accept" recommendation.

Reviewer #1: All comments have been addressed

Reviewer #2: All comments have been addressed

2. Is the manuscript technically sound, and do the data support the conclusions?

Reviewer #1: Yes

Reviewer #2: Yes

3. Has the statistical analysis been performed appropriately and rigorously? 

Reviewer #1: N/A

Reviewer #2: Yes

4. Have the authors made all data underlying the findings in their manuscript fully available?

Reviewer #1: Yes

Reviewer #2: Yes

5. Is the manuscript presented in an intelligible fashion and written in standard English?

Reviewer #1: Yes

Reviewer #2: Yes

6. Review Comments to the Author

Reviewer #1: Thank you for addressing all my concerns. The revised paper reads well and I have no further comments to make.

Reviewer #2: The Occupational Therapy Education research work is extensive and quite comprehensive, this is impressive manuscript. I accept the manuscript.

7. PLOS authors have the option to publish the peer review history of their article (what does this mean?). If published, this will include your full peer review and any attached files.

Reviewer #1: No

Reviewer #2: **Yes: **Assistant Professor Dr. Supat Chupradit

Department of Occupational Therapy, Faculty of Associated Medical Sciences, Chiang Mai University, Chiang Mai, 50200, Thailand.

Email: supat.c@cmu.ac.th

ORCID https://orcid.org/0000-0002-8596-2991

---

## [Editor Report · Acceptance letter]

10 Dec 2020

PONE-D-20-09978R1 

Research-based occupational therapy education: an exploration of students` and faculty members` experiences and perceptions 

Dear Dr. Helgøy:

I'm pleased to inform you that your manuscript has been deemed suitable for publication in PLOS ONE. Congratulations! Your manuscript is now with our production department. 

Kind regards, 

on behalf of

Dr. Sara Rubinelli 

Academic Editor

PLOS ONE